# A SINS/DVL Integrated Positioning System through Filtering Gain Compensation Adaptive Filtering

**DOI:** 10.3390/s19204576

**Published:** 2019-10-21

**Authors:** Xiaozhen Yan, Yipeng Yang, Qinghua Luo, Yunsai Chen, Cong Hu

**Affiliations:** 1School of Information Science and Engineering, Harbin Institute of Technology, Weihai 264209, China; 2Shandong Institute of Shipbuilding Technology, Weihai 264209, China; 3Guangxi Key Laboratory of Automatic Detecting Technology and Instruments, Guilin University of Electronic Technology, Guilin 541004, China; yiqi@guet.edu.cn; 4The 712th Institute of China Shipbuilding Industry Corporation, Wuhan 430064, China; 5Department of Technology, China National Deep Sea Center, Qingdao 266237, China; cys@ndsc.org.cn

**Keywords:** integrated positioning, DVL, filtering gain compensation, adaptive filter

## Abstract

Because of the complex task environment, long working distance, and random drift of the gyro, the positioning error gradually diverges with time in the design of a strapdown inertial navigation system (SINS)/Doppler velocity log (DVL) integrated positioning system. The use of velocity information in the DVL system cannot completely suppress the divergence of the SINS navigation error, which will result in low positioning accuracy and instability. To address this problem, this paper proposes a SINS/DVL integrated positioning system based on a filtering gain compensation adaptive filtering technology that considers the source of error in SINS and the mechanism that influences the positioning results. In the integrated positioning system, an organic combination of a filtering gain compensation adaptive filter and a filtering gain compensation strong tracking filter is explored to fuse position information to obtain higher accuracy and a more stable positioning result. Firstly, the system selects the indirect filtering method and uses the integrated positioning error to model the navigation parameters of the system. Then, a filtering gain compensation adaptive filtering method is developed by using the filtering gain compensation algorithm based on the error statistics of the positioning parameters. The positioning parameters of the system are filtered and information on errors in the navigation parameters is obtained. Finally, integrated with the positioning parameter error information, the positioning parameters of the system are solved, and high-precision positioning results are obtained to accurately position autonomous underwater vehicles (AUVs). The simulation results show that the SINS/DVL integrated positioning method, based on the filtering gain compensation adaptive filtering technology, can effectively enhance the positioning accuracy.

## 1. Introduction

Autonomous underwater vehicles (AUVs) play an important role during the exploration, development, and utilization of ocean resources in the military and national economy [1,2,3]. Examples are underwater sensing, observation of sea-bottom landforms, detection of specific objects, and submarine cable patrol [4,5,6]. During these operations, AUVs must know their own location and where to go, so underwater positioning technology is critical in order for underwater vehicles to safely and effectively perform their missions. However, the traditional positioning methods that are used on land cannot be used underwater and, because of the rapid attenuation of electromagnetic waves in water, the global satellite navigation system that is commonly used on land cannot be used underwater, so it is a challenge to position AUVs [7,8].

Underwater positioning methods can be classified into three groups [9,10]: dead reckoning and inertial navigation; acoustic navigation; and geophysical navigation. These positioning methods can be utilized independently or in combination to achieve integrated navigation and improve navigation accuracy. In integrated underwater navigation, the methods that are currently popular integrate the positioning of the strapdown inertial navigation system (SINS) and the Doppler velocity log (DVL) [11,12,13,14,15,16]. Ever since the successful application of this technology in the MARIDAN AUV in Denmark at the beginning of this century [13], after more than 10 years of development, it has achieved fruitful research results in several key technologies. However, due to the complex working environment, long working distances, random drift of the gyro, gradual divergence in the positioning error over time, and uncertain factors, the positioning stability and accuracy are limited and cannot meet the requirements of many marine applications. Many research works have been conducted to improve underwater positioning performance. As the positioning error resources are not considered, these improvements do not show promise. In this paper, considering positioning error resources and their impact on the positioning result, we propose a SINS/DVL integrated positioning system based on filtering gain compensation adaptive filtering technology.

In this paper, we make the following contributions:(1)For the fusion of positioning information, we present a general framework for an integrated positioning fusion system based on adaptive Kalman filtering. It provides a general reference for information fusion methods.(2)We take full advantage of strong tracking filtering and adaptive filtering to propose and utilize an organic combination of a filtering gain compensation adaptive filter and a filtering gain compensation strong tracking filter to fuse the positioning information to obtain higher accuracy and a more stable positioning result.(3)The proposed SINS/DVL integrated positioning system based on a filtering gain compensation adaptive filter can enhance underwater positioning performance and also provides reference value for other information fusion systems.

## 2. Related Works

Different types of positioning sub-systems have been integrated regarding their respective advantages and disadvantages by appropriate methods, in order for integrated positioning systems to offer complementary advantages in performance and provide better positioning performance than any single type of positioning system [1]. At present, the mainstream integrated methods include inertial navigation and Global Positioning System (GPS), DVL, and astronomic navigation. All of these positioning methods and other Kalman-filtering-related technologies have been explored in positioning information fusion [17]. However, some issues remain with the existing integrated positioning methods. When using Inertial Navigation System (INS)/GPS integrated navigation, the satellite signal can easily be occluded and distorted, the autonomy and concealment are poor, and its correct use is subject to human defects. Although INS/GPS integrated navigation has the advantage of high navigation accuracy after combination, the above drawbacks also lead to a great reduction in the reliability and application range of this combination [1]. The combination of astronomy/inertia is subject to such conditions as meteorological conditions and signal distortion, which limits the application of this integrated method [18]. Compared with the above two combinations, Doppler/inertial integrated positioning has unique advantages. The combination of the two methods has strong autonomy and concealment [19]. It can operate autonomously without external signal inputs, and the reliability is very high [20,21]. However, the amount of position information will decrease as the working time increases, which is the disadvantage of this combination method [22].

As inertial positioning systems have strong concealment and autonomy under water, they comply with the requirements of most modern underwater vehicles. Additionally, when an acoustic signal propagates in seawater, the signal attenuation decreases and the propagation distance increases. A combination of positioning systems is more widely used. The Kalman filter (KF) is a well-known powerful processing method, and has been well studied. In 1998, Savkin et al. presented a robust state estimation approach for a class of uncertain discrete-time systems with a deterministic description of noise and uncertainty. In this method, a recursive scheme was utilized to construct an ellipsoidal state estimation set of all states consistent with the measured output. And the result is promising [23]. In 2001, considering the uncertain parameters, [24] proposed a new approach to finite-horizon guaranteed state prediction for discrete-time systems affected by bounded noise and unknown-but-bounded parameter uncertainty. Wang et al. presented an adaptive robust Kalman filtering algorithm. The algorithm explores a one-step predictor-corrector structure to minimize the mean square estimation error, so as to address the estimation problem the estimation problem that arises in linear time-varying systems with stochastic parametric uncertainties in 2002 [25]. To guarantee an optimized upper bound on the state estimation error covariance despite the model uncertainties as well as the linearization errors, Kai et al. presented a novel robust extended Kalman filter (REKF) for discrete-time systems with stochastic uncertainties in 2009 [26]. In 2015, Gao et al. proposed an adaptive KF with a recursive noise estimator based on a maximum posterior estimation and one-step smoothing filtering. Adaptive filtering provides good noise parameters for a KF. The new SINS/DVL system can handle violent maneuvers and complex sea conditions very well [27]. In 2018, aiming at the problem of the low precision of the conventional filtering algorithm in the strap down inertial navigation/Doppler integrated positioning system, Guo proposed a square root unscented information filter, which reduces the computational complexity of the system while obtaining higher positioning accuracy. The results show that the tracking accuracy of the integrated positioning system is much higher than that of the SINS alone [28]. Different from previous works, in this paper, considering the parameter uncertainties and model uncertainties, we take full advantage of an organic fusion of strong tracking filtering and Sage–Husa adaptive filtering to achieve robust Kalman filtering for the SINS/DVL integrated positioning system. What is more important is that, we utilize an exponent filtering gain compensation strategy to quickly track the system state in real time, minimize the state estimation error and reduce the speed of the error accumulation. More specifically, when the initial motion state changes significantly, we increase the filter gain with an exponent compensation to enhance the effect of the new information according to the obtained observation value information. After the system is stabilized, we decrease the filter gain to ensure the optimal filtering of the system.

In the SINS/DVL integrated positioning system, the SINS has a fully autonomous navigation capability and meets the concealment requirements of underwater vehicles, so it is widely used in related fields. However, due to an underwater environment’s complexity and random drift of the gyro, using only a SINS will cause the navigation error to gradually diverge over time. Using the Doppler effect, DVL calculates the velocity of the carrier relative to the ground through an echo that transmits the beam obliquely below the carrier, providing high-precision velocity information. Therefore, SINS and DVL were integrated to form a SINS/DVL integrated navigation system. However, information on the speed of the DVL’s output can only suppress the divergence of the SINS positioning error to a limited extent. In this study, we consider the positioning parameter filtering problem in the SINS/DVL integrated positioning system and enhance the system’s positioning accuracy and stability.

## 3. SINS/DVL Integrated Positioning System Based on a Filtering Gain Compensation Adaptive Kalman Filter

In this section, we first illustrate the framework for the SINS/DVL integrated navigation system based on a filtering gain compensation adaptive Kalman filter. Then, we describe each sub-module and its implementation.

### 3.1. Framework for the SINS/DVL Integrated Positioning System

The framework for the SINS/DVL integrated positioning system is shown in Figure 1. It is composed of three sub-systems: the *Strapdown Inertial Navigation Subsystem and Error Analysis* sub-module, the *Doppler Velocity Log Subsystem and Error Analysis* sub-module, and the *SINS/DVL integrated filter* sub-module.

The *Strapdown Inertial Navigation system and Error Analysis* sub-module provides pure inertial positioning results to the SINS/DVL filter. The inertial navigation system error is used to estimate the navigation parameters in the mechanical programming of the inertial navigation system as feedback correction. In practical applications, feedback correction is generally used; however, Kalman filtering takes time to stabilize, and the error is very large at first, which leads to a decrease in system accuracy and is not suitable for feedback. To address these issues, we use a combination of output correction and feedback correction.

The *Doppler Velocity Log Subsystem and Error Analysis* sub-module provides high-precision velocity information and performs an error analysis in DVL. In this system, we utilized the difference in value between the solution velocity of the strap down inertial navigation system and the velocity of the Doppler velocity log as the observation of the SINS/DVL filter.

The *SINS/DVL integrated filter* sub-module is the positioning information fusion center. It is composed of a SINS/DVL integrated navigation error model and filtering gain compensation-based adaptive filtering. The SINS/DVL integrated navigation error model constructs the system state variable, the system state equation, and the system measurement equation. The filtering gain compensation-based adaptive filter performs data fusion and filtering. It is an organic combination of an adaptive filtering and a strong tracking Kalman filtering. We combine the inverse-free matrix with the improved adaptive filter, so a high positioning accuracy with stability and efficiency can be obtained.

The SINS/DVL integrated positioning system is suitable for a remote AUV underwater navigation. When we describe the system’s dynamic characteristics and measurement equations, we divide the integrated navigation filtering methods into direct and indirect methods. Compared with the direct method, the indirect method has the characteristics of relatively high complexity and higher accuracy. We use the indirect filtering to obtain higher accuracy. The indirect filtering has output correction and feedback correction. Regarding the output correction, we use the inertial navigation system’s error estimation to correct the navigation parameters of the inertial navigation system.

### 3.2. Strapdown Inertial Navigation Subsystem and Error Analysis

The strap down inertial navigation system consists of a three-axis accelerometer assembly and three orthogonal rate gyros. Without a solid platform, both the accelerometer and the inertial gyros are mounted on the target. The cost is lower, the structure is simpler, and the space utilization rate is higher; however, the calculation process is more complicated. To aid in computing, another important part of the Strapdown Inertial Navigation System is its computer.

During the movement of the target, the installed gyroscope obtains the angular velocity of the aircraft relative to the inertial reference frame. Taking this into account, the transformation matrix of the coordinate system that corresponds to the target coordinate system can be calculated in order to obtain the three-axis accelerometer. The acceleration variable is fed back into the positioning coordinate system, and the specific navigation information is obtained through the final operation of the computer.

During the navigation process, an underwater vehicle is equipped with an accelerometer and a gyroscope in the northeast direction of the navigation coordinate system. Let the acceleration of the carrier in the north direction be *a_n_*. Similarly, the east acceleration and the vertical acceleration are recorded as *a_e_* and *a_u_*, respectively. With the passage of time, the acceleration is integrated with time, and the velocity components in the three directions of the aircraft can be calculated:(1)$$\{\begin{array}{c} v_{e}(t_{k})=v_{v}(t_{0})+\int_{t_{0}}^{t_{k}}a_{e}dt\\ v_{n}(t_{k})=v_{v}(t_{0})+\int_{t_{0}}^{t_{k}}a_{n}dt\\ v_{u}(t_{k})=v_{v}(t_{0})+\int_{t_{0}}^{t_{k}}a_{u}dt \end{array}$$

In the same way, to obtain the latitude, longitude, and depth of the aircraft on the earth, we can also obtain Equation (2) from Equation (1):(2)$$\{\begin{array}{c} \lambda =\lambda_{0}+\int_{t_{0}}^{t_{k}}\dot{\lambda }dt\\ j=j_{0}+\int_{t_{0}}^{t_{k}}\dot{j}dt\\ d=d_{0}+\int_{t_{0}}^{t_{k}}\dot{d}dt \end{array}$$
where $\lambda_{0}$, $j_{0}$ and $d_{0}$ are the initial positions of the warp, weft, and depth of the aircraft, respectively, and $\dot{\lambda }$, $\dot{j}$ and $\dot{d}$ are the rates of change of the warp, latitude, and depth, respectively. The speed can be tracked in real time by the SINS/DVL combined positioning system, so the rate of change of warp, latitude, and depth can be extrapolated from speed as follows:(3)$$\{\begin{array}{l} \dot{\lambda }=\frac{v_{e}}{(N+d)\cos\varphi }\\ \dot{j}=\frac{v_{n}}{M+d}\\ \dot{d}=v_{u} \end{array}.$$

In the formula, in order to simplify the calculation, we approximate the Earth as a sphere with radius *R*. Then, *M* = *N* = *R*.

The inertial navigation system requires a series of reference coordinate systems to define the position and state of motion of the aircraft. When the attitude calculation is performed on the mathematical platform, the reference system is converted. The common coordinate system of the SINS/DVL integrated navigation system is the local horizontal coordinate system.

The horizontal coordinate system is the coordinate system of the relative geoid plane, which is also known as the geographic coordinate system. The origin is set at the center of the aircraft, and is also the projection of the origin of the inertial platform on the geoid level. The direction axes of the horizontal coordinate system point to the north, the east, and the sky. The e-axis points along the east direction of the reference ellipsoid circle (*x*-axis), the n-axis points along the meridian circle to the north direction (*y*-axis), and the u-axis points along the normal outside of the ellipsoid to the sky (*z*-axis). Sometimes the x, y, and z axes are referred to as the east–north–sky axis. Because of their widespread use in the navigation field, the horizontal coordinate system is also called the navigation coordinate system.

The transformation matrix of the carrier coordinate system relative to the local horizontal coordinate system is as follows:(4)$$R_{b}^{L}=\left[\begin{array}{ccc} \cos r\cos y-\sin r\sin y\sin p & -\sin y\cos p & \cos y\sin r+\sin y\sin p\cos r\\ \cos r\sin y+\sin r\cos y\sin p & \cos y\cos p & \sin y\sin r-\cos y\sin p\cos r\\ -\cos p\sin r & \sin p & \cos p\cos r \end{array}\right]$$
where (*r*, *p*, *y*) are the roll angle, the pitch angle, and the heading angle of the underwater vehicle, respectively. We also call $R_{}{}_{b}^{L}$ the attitude matrix.

The error in the strap down inertial navigation system is mainly caused by the following four situations:
*Errors caused by inertial sensors*: Inertial sensors, such as accelerometers or gyroscopes, may cause errors due to processing, assembly processes, or principal problems. This type of error is called systematic error. In the Strapdown Inertial Navigation/Doppler Velocity Logger integrated positioning system, such errors account for 90% of the total error.*Errors in the computer operation process*: The Strapdown Inertial Navigation System outputs the acceleration and attitude angles of the gyroscope and accelerometer, and the rest of the computational work is done by the computer. The calculation error includes the error in the Strapdown Inertial Navigation System and the error in the Kalman filter.*Errors caused by mathematical models*: When establishing the system model, we approximate the earth as a sphere in the calculation process. When using Newton’s second law as the inertial motion equation, and the speed of the aircraft is very high, the description of the aircraft’s motion will be insufficiently accurate.*Errors caused by the initial alignment*: These errors can be divided into the following:
(1)Installation errors. Gyros and accelerometers can cause errors during installation.(2)Initial condition errors. Initial condition errors include heading, attitude and position, and zero errors in the gyroscope and accelerometer.(3)Calculation errors. The calculation errors of the navigation system are also called the calculation errors of the mathematical platform.(4)Instrument errors. This refers mainly to the errors in the drift that is generated by the gyroscope and the accelerometer and the error in the angular rate meter.

In the SINS/DVL integrated filter’s design, we consider these error sources and take steps to minimize their negative effect in order to obtain a stable and accurate positioning result.

### 3.3. Doppler Velocity Log Subsystem and Error Analysis

The Doppler log operates by the phenomenon of the aircraft generating a Doppler shift during the ultrasonic wave’s movement to the sea floor. The Doppler effect refers to a phenomenon in which the frequency of the acoustic wave that is received by the observing object (or the observed object) and the frequency of the emitted sound wave are different when the observing object (or the observed object) moves relative to the reference medium. The Doppler log provides a standard speed relative to the acoustic wave source in the carrier coordinate system during normal operation. The standard speed that is provided includes not only information on the forward motion of the aircraft relative to the submarine base, but also the current aircraft, the drift angle of the heading, and information on the lateral displacement velocity that is caused by the influence of the ocean current relative to the seabed base.

The DVL receiver and transmitter are typically fixed on the same AUV, with the AUV moving at a speed *ν* relative to the seafloor reflection point. The acoustic frequency of the DVL is *f*. The acoustic frequency received by the reflection point at the bottom of the sea is *f’*, and the acoustic beam propagates in the water at the speed of *ν*_0_. According to the Doppler shift formula, the following formula is given:(5)$$f^{\prime }=\frac{v}{v-v_{0}}f.$$

The frequency of the received acoustic beam from the seafloor reflection is *f′*, and the sound frequency of the DVL that is captured by the underwater vehicle during the movement is *f″*, which is:(6)$$f^{\prime \prime }=\frac{v+v_{0}}{v}f^{\prime }=\frac{v+v_{0}}{v-v_{0}}f.$$

According to (6), the Doppler shift is:(7)$$f_{d}=f^{\prime \prime }-f=\frac{2v_{0}}{v-v_{0}}f.$$

After integrating the above, we can obtain the speed formula of the aircraft as follows:(8)$$v=(1+\frac{2f}{f_{d}})v_{0}.$$

The Doppler log not only measures the speed of the underwater vehicle’s forward and reverse motion, but also measures the speed of the underwater vehicle’s lateral displacement. The navigation system that is realized by Doppler log meets the requirements for remote underwater navigation without relying on external equipment for independent navigation work and is not affected by the failure of navigation base stations in the air and land. The impact of the visual visibility can also be completely ignored because of Nano positioning. At the same time, the Doppler sonar can provide a more accurate measurement of speed in a shallow water environment when the aircraft is in low-speed operation. The lowest measurable speed is 0.005 m/s. The shallowest water depth is between 0.3 and 0.5 m. In many cases, autonomous vehicles navigate at a relatively low speed. Under the premise of low-speed navigation, the Doppler log can also maintain sufficient sensitivity for positioning work. Although the Doppler velocimeter uses a sound wave signal to perform navigation speed measurements, since the angle of injection into the sea floor is steep, it still maintains good concealment. The anti-interference is strong, the sensitivity is high, and the speed measurement’s accuracy is higher. However, the Doppler log requires an external vertical reference and heading information during actual navigation work, which is why the Doppler log and the strap down inertial navigation system were integrated. Since the SINS/DVL integrated positioning system belongs to an externally independent dead reckoning system, positioning errors will also accumulate over time. If a remote autonomous underwater vehicle based on the Strap Down Inertial Navigation/Doppler Logger Integrated Positioning System is used to perform a task, it is necessary to periodically perform positioning correction by means of the “Snorkeling–Surface Correction–Submarine” GPS positioning system. The calibration period is long, so it does not affect the long-lasting endurance and navigation independence of the integrated positioning system.

According to the working principle of the DVL, the DVL can obtain information on the speed and drift angle of an AUV relative to the seabed. The underwater vehicle speed ${\tilde{V}}_{d}$ and drift angle $\tilde{\delta }$ error equations are as follows:(9)$$\tilde{\delta }=\delta +\delta \Delta$$

(10)$${\tilde{V}}_{d}=(1+\delta C)V_{d}+\delta V_{d}.$$

In the formulae, δΔ1, $\delta V_{d}$, and δC are the bias angle error, the velocity error, and the coefficient error of the AUV, respectively. Based on the actual operating environment of the Doppler log, we take $\delta V_{d}$ and δΔ as first-order Markov processes. δC is a constant. Then, we can obtain:(11)$$\delta {\tilde{V}}_{d}=-\beta_{d}\delta V_{d}+\omega_{d}$$
(12)$$\delta \dot{\Delta }=-\beta_{d}\delta \Delta +\omega_{\Delta }$$
(13)$$\delta \dot{C}=0$$
where $\beta_{d}^{-1}$ and $\beta_{\Delta }^{-1}$ are the correlation times of the velocity offset error and the drift angle error, respectively, and $\omega_{d}$ and $\omega_{\Delta }$ are white noise.

### 3.4. SINS/DVL Integrated Adaptive Filter

In this section, we first establish the integrated navigation error model. Then, we describe the filtering gain compensation adaptive Kalman filter algorithm that we use to process and fuse the positioning information.

#### 3.4.1. Establishment of the Navigation Error Model

The error model of the integrated navigation system is composed of a strapdown inertial navigation system error model and a Doppler log error model. A long distance underwater vehicle has a long working distance. Based on the above characteristics, we chose the positioning error, velocity error, misalignment angle, gyro drift, speed offset error, drift angle error, and scale factor error as the state quantities for the filtering. The gyro drift model is described by a first-order Markov model. For the Doppler log, it provides the system with the velocity and drift angle of the carrier relative to the seabed. The measurement error is divided into the drift angle error, the velocity error and the scale factor error, which are also defined by the first-order Markov model.

The system state variable in the error model is $X={\left[\delta v_{E}\;\delta v_{N}\;\alpha \;\beta \;\gamma \;\delta L\;\delta \lambda \;\varepsilon_{E}\;\varepsilon_{N}\;\varepsilon_{U}\;\delta v_{d}\;\delta \Delta \;\delta C\right]}^{T}$, where $\delta v_{E}$ and $\delta v_{N}$ are the east and north speed error, respectively, α, β, and γ is the platform misalignment angle, δL and δλ are the latitude and longitude error, respectively, $\varepsilon_{E}$, $\varepsilon_{N}$, and $\varepsilon_{U}$ are the gyro drift, $\delta v_{d}$ is the measuring velocity offset error for the Doppler log, δΔ is the bias angle error, and δC is the scale factor error.

(1) *System State Equation*

The east and north speed error formulae are given as follows:(14)$$\delta v_{E}=\frac{v_{N}}{R_{N}}\cdot \tan L\cdot \delta v_{E}+(2\Omega \sin L+\frac{v_{E}}{R_{N}}\cdot \tan L)\delta v_{N}+(2\Omega \cos Lv_{N}+\frac{v_{N}v_{E}}{R_{N}}\cdot \sec^{2}L)\delta L-\beta g+\Delta a_{E}$$
(15)$$\delta v_{N}=(2\Omega \sin L+\frac{v_{E}}{R_{N}}\cdot \tan L)\delta v_{E}+(2\Omega \cos Lv_{E}+\frac{v_{E}^{2}}{R_{N}}\cdot \sec^{2}L)\delta L+\alpha g+\Delta a_{N}.$$

The platform misalignment angles are defined as follows:(16)$$\alpha =-\frac{\delta v_{N}}{R}-\gamma (\Omega \cos L+\frac{v_{E}}{R})+\beta (\Omega \sin L+\frac{v_{E}}{R}\tan L)+\varepsilon_{E}$$
(17)$$\beta =-\alpha (\Omega \sin L+\frac{v_{E}}{R}\tan L)-\delta L\Omega \sin L+\frac{\delta v_{E}}{R}-\gamma \frac{v_{E}}{R}+\varepsilon_{N}$$
(18)$$\gamma =\delta L(\Omega \cos L+\frac{v_{E}}{R}\sec^{2}L)+\frac{\delta v_{E}}{R}\tan L+\beta \frac{v_{N}}{R}+\alpha (\Omega \cos L+\frac{v_{E}}{R})+\varepsilon_{U}.$$

The positioning error is denoted as:(19)$$\delta L=\frac{\delta v_{N}}{R}$$
(20)$$\delta \lambda =\frac{\delta v_{E}}{R}\sec L+\frac{v_{E}}{R}\tan L\sec L\delta L.$$

The gyro drift is defined as follows:(21)$$\varepsilon_{E}=-\beta_{E}\varepsilon_{E}+w_{E}$$
(22)$$\varepsilon_{N}=-\beta_{N}\varepsilon_{N}+w_{N}$$
(23)$$\varepsilon_{U}=-\beta_{U}\varepsilon_{U}+w_{U}$$
where $\beta_{E}^{-1}$, $\beta_{N}^{-1}$, and $\beta_{U}^{-1}$ are for the gyroscope in the east, the gyroscope in the north, and the direction of the error related to time.

The speed, drift angle, and scale error of the Doppler log are given as follows:(24)$$\delta v_{d}=-\beta_{d}\delta v_{d}+w_{d}$$

(25)$$\delta \Delta =-\beta_{\Delta }\delta \Delta +w_{\Delta }$$

(26)δC=0.

The state equation is described as:(27)$${\dot{X}}_{SINS/DVL}=F_{SINS/DVL}X_{SINS/DVL}+G_{SINS/DVL}W_{SINS/DVL}$$

In the formula:(28)$$W_{SINS}={\left[\begin{array}{ccccccc} 0 & 0 & a_{E} & a_{N} & 0 & 0 & 0 \end{array}\;w_{E}\;w_{N}\;w_{U}\;w_{d}\;w_{\Delta }\;0\right]}^{T}.$$

We establish a state transfer matrix according to Equations (14)–(28):(29)$$F_{SINS/DVL}=\left[\begin{array}{cc} F_{7\times 7} & \begin{array}{c} \begin{array}{cc} 0_{4\times 3} & 0_{4\times 3} \end{array}\\ \begin{array}{cc} I_{3\times 3} & 0_{3\times 3} \end{array} \end{array}\\ 0_{6\times 7} & F_{6\times 6} \end{array}\right]$$
where: (30)$$F_{7\times 7}=\left[\begin{array}{lllllll} 0 & 0 & 0 & F_{14} & 0 & 0 & 0\\ F_{21} & 0 & F_{23} & 0 & 0 & 0 & 0\\ F_{31} & 0 & F_{33} & F_{34} & 0 & -g & 0\\ F_{41} & 0 & F_{43} & 0 & g & 0 & 0\\ 0 & 0 & 0 & F_{54} & 0 & F_{56} & F_{57}\\ F_{61} & 0 & F_{63} & 0 & F_{65} & 0 & F_{67}\\ F_{71} & 0 & F_{73} & 0 & F_{75} & F_{76} & 0 \end{array}\right]$$

In matrix $F_{7\times 7}$, $F_{14}=\frac{1}{R}$, $F_{21}=\frac{V_{E}}{R}\tan L\sec L$, $F_{23}=\frac{\sec L}{R}$, $F_{31}=(2\Omega \cos L+\frac{V_{E}}{R}\sec^{2}L)V_{N}$, $F_{33}=\frac{V_{N}}{R}\tan L$, $F_{34}=2\Omega \sin L+\frac{V_{E}}{R}\tan L$, $F_{36}=-g$, $F_{41}=(2\Omega \cos L+\frac{V_{E}}{R}\sec^{2}L)V_{E}$, $F_{43}=-(2\Omega \sin L+\frac{V_{E}}{R}\sec^{2}L)V_{E}$, $F_{45}=g$, $F_{54}=-\frac{1}{R}$, $F_{56}=\Omega \sin L+\frac{V_{E}}{R}\tan L$, $F_{57}=-(\Omega \cos L+\frac{V_{E}}{R})$, $F_{61}=-\Omega \sin L$, $F_{63}=\frac{1}{R}$, $F_{65}=-(\Omega \sin L+\frac{V_{E}}{R}\tan L)$, $F_{67}=\frac{V_{N}}{R}$, $F_{71}=\Omega \cos L+\frac{V_{E}}{R}\sec^{2}L$, $F_{73}=\frac{\tan L}{R}$, $F_{75}=\Omega \cos L+\frac{V_{E}}{R}$, $F_{76}=\frac{V_{E}}{R}$.

(31)$$F_{6\times 6}=\left[\begin{array}{llllll} -\beta_{E} & 0 & 0 & 0 & 0 & 0\\ 0 & -\beta_{N} & 0 & 0 & 0 & 0\\ 0 & 0 & -\beta_{U} & 0 & 0 & 0\\ 0 & 0 & 0 & -\beta_{d} & 0 & 0\\ 0 & 0 & 0 & 0 & -\beta_{\Delta } & 0\\ 0 & 0 & 0 & 0 & 0 & 0 \end{array}\right]$$

(2) *System Measurement Equation*

Since we use the difference in value between the solution speed of the strap down inertial navigation system and the speed of the Doppler log as the observation of the measurement equation, the system measurement equation is:(32)$$Z_{SINS/DVL}=\left[\begin{array}{c} \delta v_{E}-\delta v_{dE}\\ \delta v_{N}-\delta v_{dN} \end{array}\right]=H_{SINS/DVL}X_{SINS/DVL}+V_{SINS/DVL}$$

In Equation (32):(33)$$H_{SINS/DVL}=\left[\begin{array}{lllllllllllll} 0 & 0 & 1 & 0 & 0 & 0 & -v_{N} & 0 & 0 & 0 & -\sin K_{d} & -v_{N} & -v_{E}\\ 0 & 0 & 0 & 1 & 0 & 0 & v_{E} & 0 & 0 & 0 & -\cos K_{d} & v_{E} & -v_{N} \end{array}\right].$$
(34)$$V_{SINS/DVL}={\left[\begin{array}{cc} v_{E} & v_{N} \end{array}\right]}^{T}$$


Here, the system noise variance matrix is:(35)$$Q=diag(R_{\delta V_{E}}\;{}_{}R_{\delta V_{N}}\;{}_{}R_{\alpha }\;{}_{}R_{\beta }\;{}_{}\;R_{\gamma }\;{}_{}\;R_{\delta L}\;{}_{}\;R_{\delta \lambda }\;{}_{}\;R_{\varepsilon E}\;{}_{}\;R_{\varepsilon N}\;{}_{}\;R_{\varepsilon U}\;{}_{}\;R_{\delta Vd}\;{}_{}R_{\delta \Delta }\;{}_{}R_{\delta C}).$$

The measurement noise variance matrix is:(36)$$R=diag(\begin{array}{cc} R_{V_{N}} & R_{V_{E}} \end{array}).$$

#### 3.4.2. Filtering Gain Compensation-Based Adaptive Kalman Filter

(1) Workflow of the Filtering Gain Compensation Adaptive Filter

Figure 2 shows the workflow of the filtering gain compensation adaptive filtering algorithm. The filtering gain compensation algorithm was combined with the filtering gain compensation adaptive filter and the filtering gain compensation strong tracking Kalman filter. After filter initialization and parameter setting, we calculate the AUV’s state of motion using the Kalman filter, and judge whether the filter procedure is convergent or not, i.e., we determine whether the estimated error value is much larger than the error’s expected value or not. If it is, we use the filtering gain compensation adaptive Kalman filter to filter the position information. Otherwise, we utilize the filtering gain compensation strong tracking Kalman filter. Lastly, if the time period ends, we come to the procedure’s end. Otherwise, we perform the filter gain compensation adaptive filtering iteratively.

(2) Filtering Gain Compensation

When a Kalman filter is used for the long-range underwater dynamic tracking of an AUV, if the external interference in the underwater current or the speed and course of the underwater vehicle experience a sudden change, the estimated value from the filtering process will be slightly delayed when tracking the real state of the underwater vehicle. This will reduce the navigation accuracy and quality.

In the process of Kalman filtering, filtering gain and the one-step predicted value are the main factors that lead to the failure of state estimation to track the real state of the vehicle in time. For state estimation, the filtered gain $K_{k}$ represents the action strength of the information residual. If the $K_{k}$ value is large, the correction of the estimated value at the next moment will be large. In contrast, if $K_{k}$ is small, the predicted value at the current moment will be more accurate. Therefore, the degree to which the actual state of motion at the moment *k*+1 coincides with the estimation of the optimal state of motion is directly related to the correction of the state estimation by the filtering magnification factor $K_{k}$.

When the Kalman filter and the positioning system are stable, the real state of the system corresponds to the one-step predicted state value. Due to the small size of the error, the correction of ${\hat{X}}_{k,k-1}$ is small. However, due to the changes in the system state during the filtering process, the estimate fails to follow the change in the target’s maneuvers. The reason for this is that, in the estimation of the next moment, the system is always judged according to its previous state, and the state transition matrix at the previous moment is used to calculate the predicted state variables at the next moment, which will inevitably lead to the gradual accumulation of errors.

In order to quickly track the system state in real time and reduce the speed of error accumulation, we can improve the filter gain $K_{k}$.

We can obtain the required observation values from the sensors (such as the velocity as displayed by the Doppler log or the attitude angle and acceleration as displayed by the SINS inertial navigation system). The error caused by the correlation between the observed value and the system is relatively small. We can increase the filter gain value $K_{k}$ and enhance the effect of the new information according to the obtained observation values. In addition, the covariance error matrix $P_{k}$ in the filtering process is also related to the filtering gain; an increase in $P_{k}$ will result in an increase in $K_{k}$. Since $P_{k}$ is the covariance matrix of the error, $P_{k}$ is also the embodiment of the filtering accuracy. If $K_{k}$ is appropriately increased, the filtering correction will be large and the error covariance matrix $P_{k}$ will also be large.

To let the improved filter gain K be $K_{n}$, and, on the basis of the original gain, we add to K a supplementary term $K_{g}$, as shown in Equation (37):(37)$$K_{n}(k)=K(k)+K_{g}(k).$$

In Equation (37), K(k) is still calculated in the traditional way, and the corresponding filtering equation is updated as follows:(38)$$\hat{K}(k+1)=\hat{K}(k+1/k)+\hat{K}(k+1)[\hat{Z}(k+1)-\hat{Z}(k+1/k).$$

In order to make the stabilized system filter optimally, it is necessary for $K_{n}$ to increase the tracking performance over time when the initial state of motion changes significantly. After the system is stabilized, $K_{n}$ can be reduced to $K_{g}$ to ensure that the system is optimally filtered, as shown in (39):(39)$$K_{g}=K_{z}\lambda^{\left(k-k_{0}\right)}.$$

λ must be less than 1 in order to ensure that the filter gain of the vehicle increases rapidly at the moment of an abrupt transition and then decreases rapidly in order to ensure that the optimal filter conditions are satisfied. The size of λ also reflects the speed of the change in the filter gain. $k_{0}$ denotes the time when the navigation system detects a sudden change in the state of motion of the vehicle, and $P_{k}$ represents the deviation in the error of the state variable in the filtering process. In the case of a sudden change at $k_{0}$, a large $P_{k}$ will be selected because the tracking response of the system is not timely and the error is large. The expressions of the covariance matrix and the filter gain at this time are as shown in (40) and (41):(40)$$P_{z}(k_{0})=\Phi (k_{0},k_{0}-1)P_{z}\Phi^{T}(k_{0},k_{0}-1)$$

(41)$$K=H(k_{0})H^{T}(k_{0}){\left[H(k_{0})P_{z}(k_{0})H^{T}(k_{0})\right]}^{-1}.$$

(3) Adaptive Kalman Filter 

To improve the stability, minimize the error, and restrain the divergence of the dynamic filtering, the Sage–Husa adaptive filter add statistical characteristics of noise to the observation data when estimating the observation of system state at the next moment. We estimate and correct the status of the observation noise and system noise over time. The discrete time linear system is defined as follows:(42)$$X_{k}=F_{k,k-1}X_{k-1}+G_{k-1}W_{k-1}$$
(43)$$Z_{k}=H_{k}X_{k}+V_{k}$$
where $W_{k}$ and $V_{k}$ are normal white noise series:(44)$$\{\begin{array}{ll} E[W_{k}]=q_{k} & E[W_{k}W_{j}^{T}]=Q_{k}d_{kj}\\ E[V_{k}]=r_{k} & E[V_{k}V_{j}^{T}]=R_{k}d_{kj}\\ E[W_{k}V_{j}^{T}]=0 & \end{array}$$

The Sage–Husa adaptive filter algorithm can be expressed as follows:(45)$${\hat{X}}_{k}={\hat{X}}_{k,k-1}+K_{k}{\tilde{Z}}_{k}$$

(46)$${\hat{X}}_{k,k-1}=F_{k,k-1}{\hat{X}}_{k-1}+{\hat{q}}_{k}$$

(47)$${\hat{Z}}_{k}=Z_{k}-H_{k}{\hat{X}}_{k,k-1}-{\hat{r}}_{k}$$

(48)$$K_{k}=P_{k,k-1}H_{k}^{T}{\left(H_{k}P_{k,k-1}H_{k}^{T}+{\hat{R}}_{k}\right)}^{-1}$$

(49)$$P_{k,k-1}=F_{k,k-1}P_{k,k-1}F_{k,k-1}^{T}+{\hat{Q}}_{k-1}$$

(50)$$P_{k}=(I-K_{k}H_{k})P_{k,k-1}.$$

Here, ${\hat{r}}_{k}$, ${\hat{R}}_{k}$, ${\hat{q}}_{k}$, and ${\hat{Q}}_{k}$ can be estimated by the time variable estimation equation:(51)$${\hat{r}}_{k+1}=(1-d_{k}){\hat{r}}_{k}+d_{k}(Z_{k+1}-H_{k+1,k}{\hat{X}}_{k+1,k})$$

(52)$${\hat{R}}_{k+1}=(1-d_{k}){\hat{R}}_{k}+d_{k}({\tilde{Z}}_{k+1}{\tilde{Z}}_{k+1}^{T}-H_{k+1,k}P_{k+1}H_{k+1}^{T})$$

(53)$${\hat{q}}_{k+1}=(1-d_{k}){\hat{R}}_{k}+d_{k}({\hat{X}}_{k+1}-F_{k+1}\hat{X})$$

(54)$${\hat{Q}}_{k+1}=(1-d_{k}){\hat{Q}}_{k}+d_{k}(K_{k+1}{\tilde{Z}}_{k+1}{\tilde{Z}}_{k+1}^{T}K_{k+1}^{T}+P_{k+1}-F_{k+1}P_{k}F_{k+1}^{T}).$$

We call $d_{k}=(1-d_{k})/(1-b^{k+1}),\;0<b<1$ the “forgetting factor. The status variables of the navigation system have many reference values, which leads to the system status variables having high dimensions. At the same time, the variance in the noise estimation value needs to be calculated during filtering. The number of required calculations is very high. So, the filter cannot track the status variable in real time and the stability of the system cannot be guaranteed.

(4) Strong Tracking Kalman Filter

In order to ensure the reliable convergence of the filter, some accuracy can be sacrificed to ensure that the filter remains stable. In strong tracking Kalman filtering, we consider the error during the modeling of the system by adding an observation noise series and a process noise series. More specifically, the pre-covariance matrix of the state estimation error is multiplied by a weighted coefficient to maintain the ability to track abrupt changes in the system state.

Suppose the weighted coefficient is $\lambda_{K+1}$. Then, the weighting process can be expressed as follows:(55)$$P_{k,k-1}=\lambda_{k-1}\Phi_{k,k-1}P_{k,k-1}\Phi_{k,k-1}^{T}+{\hat{Q}}_{k-1}$$

For the weighted coefficient algorithm:(56)$$\lambda_{k+1}=diag[\lambda_{1(k+1)},\lambda_{2(k+1)},\ldots ,\lambda_{m(k+1)}]$$
(57)$$\lambda_{i(k+1)}=\{\begin{array}{c} \alpha_{i}C_{k+1}(\alpha_{i}C_{k+1}>1)\\ 1\;\;(\alpha_{i}C_{k+1}<1) \end{array}$$
(58)$$C_{k+1}=\frac{Tr[v_{0(k+1)}-R_{k+1}-H_{k+1}Q_{k}H_{k+1}^{T}]}{\sum_{i=1}^{n}\alpha_{i}[\Phi_{k+1}P_{k+1}\Phi_{k+1}^{T}H_{k+1}^{T}H_{k+1}]}$$
(59)$$v_{0(k+1)}=\{\begin{array}{c} {\tilde{Z}}_{1}{\tilde{Z}}_{1}^{T}\;\;\;(k=0)\\ \frac{\rho v_{0(k)}+{\tilde{Z}}_{k+1}Z_{k+1}^{T}}{1+\rho }(k\geq 1,0\leq \rho <1) \end{array}$$
where $\alpha_{i}$ is the experience coefficient. When the status of the underwater vehicle changes rapidly, the error estimation ${\tilde{Z}}_{k+1}{\tilde{Z}}_{k+1}^{T}$ can lead to an increment in the error variance $v_{0(k+1)}$, and the weighted coefficient $\lambda_{K+1}$ will be larger. The ability of the filter to adapt will be strengthened when facing rapid changes in status.

## 4. Simulation Results and Analysis

Based on the filtering algorithm and the SINS/DVL integrated navigation system algorithm we described in the previous section, we verify the feasibility of and validate our proposed method in the Matlab environment, and also compare our proposed method with other related methods.

### 4.1. Simulation Setup

#### 4.1.1. Motion Model and Simulation Design

In order to verify the accuracy and stability of the filtering process, we established a motion model in which the state of motion changed with time. We assumed that the AUV’s travel time was 2000 s. First, the AUV traveled in a straight line with a north velocity of 2.57 m/s and an east velocity of 0.27 m/s for 1000 s. Then, a heading rate of change of AUV to the east was used for steering, and the course change time was 25 s. Then, the speed was maintained after steering from 1025 s to 1600 s. A heading rate of change of 3°/*s* to the north was used for turning at the time of 1600 s, and the course change time was 25 s. Finally, the AUV continued in a straightforward direction after turning. From 1000 s to 2000 s, the drive speed was 5 (2.57 m/s). The process of motion is shown in Figure 3 and Figure 4.

Based on the model of the target’s motion, we evaluated the positioning accuracy and stability of our proposed method, especially when the velocity changes.

#### 4.1.2. Evaluation Metric

We evaluated the performance of our proposed method in terms of positioning accuracy and efficiency. We also considered the stability of the positioning method when the system state changes dynamically. For positioning accuracy, we considered the eastward distance error, as the north distance error changed little during the AUV’s motion. For the positioning efficiency, we assessed the processing time of the positioning method.

#### 4.1.3. Reference Positioning Methods

We compared our proposed method with related positioning methods, including the classical Kalman filter, the strong tracking filter, the Sage–Husa adaptive Kalman filter, and the improved adaptive filter. 

(1) Classical Kalman Filter

In a complex underwater environment, the positioning accuracy and stability of this filter is very low. In the SINS/DVL integrated system, due to the instability of the components of the system itself and the uncertainties in the external environment, it is difficult to bring accurate statistical characteristics of noise to the system. With the conventional filters, stability and convergence are not guaranteed.

(2) Strong Tracking Filter

This filter sacrifices positioning accuracy to ensure the reliable convergence of the filter. In strong tracking filtering, the pre-covariance matrix of the state estimation error is multiplied by a weighted coefficient to maintain the ability to track abrupt changes in the system state.

(3) Improved Adaptive Filter

To improve the stability, minimize the error, and reduce the divergence of positioning error in dynamic filtering, the Sage–Husa adaptive filter adds statistical characteristics of noise to the observation data when estimating the system state at the next moment. The adaptive algorithm adds a calculation of the system’s statistical characteristics, which greatly increases the number of calculations and cannot guarantee the stability and convergence of a higher-order system. The adaptive Sage–Husa adaptive algorithm and the strong tracking Kalman filter have their own advantages and disadvantages. The former has high precision but a large computational burden, so it cannot guarantee real-time tracking and stability. The latter has a simple structure, high reliability, and a strong adaptive ability, but its filtering accuracy is reduced. When integrated, they give full effect to their advantages and constitute an improved Kalman filtering algorithm that not only suppresses the divergence of positioning error during filtering but also ensures the accuracy and efficiency of the filter. This method uses a convergence criterion to make a judgment about the divergence trend of the filter. If the filter does not diverge, the Sage–Husa adaptive algorithm is adopted. The strong tracking Kalman filtering algorithm is adopted when a divergence trend exists.

### 4.2. Positioning Accuracy and Stability Evaluation

We evaluated the positioning accuracy of the proposed method under the model of the AUV’s motion, and compared it with that of other related methods. We also considered the stability of the positioning system. Figure 5 shows the positioning error of these methods. Table 1 illustrates the statistical positioning error parameter of the different methods.

#### 4.2.1. Positioning Accuracy Analysis

It can be seen from Figure 5 and Table 1 that the positioning accuracy of the proposed filtering gain compensation-based adaptive filter is significantly better than that of the other related methods. More specifically, the positioning error of our proposed method is the lowest among these methods. From the viewpoint of the minimum, maximum, and peak–peak error value, the lower the peak–peak value is, the higher the positioning accuracy is. So, we can see that the filtering gain compensation-based filter performs better than the other methods. From the viewpoint of the mean positioning error, when compared to the classical, strong tracking, and improved adaptive filtering methods, the proposed method improves the positioning accuracy by 98.70%, 59.06%, and 106.46%, respectively. This result is mainly due to the Kalman filter not being able to track the system in real-time due to the sudden changes in the state of motion of the AUV at 1000 s and 1600 s, so the positioning error in the classical Kalman filter is very large. The filtering gain compensation-based filter takes advantage of the combination of the filtering gain compensation adaptive filter and the filtering gain compensation strong tracking Kalman filter. It can track the rapid changes in the AUV’s state of motion, and can better correct system errors and improve the positioning accuracy, so it has the lowest positioning error, and can also guarantee the system’s stability.

(1) Comparison with Classical Kalman Filtering

It can be seen from Figure 5 and Table 1 that the positioning accuracy of the filtering gain compensation-based filter is significantly better than that of the classical Kalman filter. The peak–peak positioning error of the proposed method is 0.2476 m. The corresponding error of the classical Kalman filter is 6.9751 m. The proposed method improves the positioning accuracy by 98.70%. This is due mainly to the classical Kalman filter being unable to track the sudden changes in the AUV’s state of motion at 1000 s and 1600 s. The filtering gain compensation-based filtering method can better correct system errors and enhances the positioning accuracy.

(2) Comparison with Strong Tracking Filtering and Improved Adaptive Filtering

From Figure 5 and Table 1, it can be seen that the positioning error of the filtering gain compensation-based filter is much lower than that of the strong tracking filter and the improved adaptive filter. More specifically, compared with the two filtering methods, the proposed method improves the positioning accuracy by 59.06%, and 106.46%, respectively. Although the two methods are able to track the sudden changes in the AUV’s state of motion at 1000 s and 1600 s, the positioning error fluctuates significantly. The proposed method utilizes the error statistics of the position parameter and filtering gain compensation-based adaptive filtering to track the sudden changes in the AUV’s state of motion. So, it produces the highest positioning accuracy.

#### 4.2.2. Positioning Stability Analysis

We evaluated the positioning stability in terms of the fluctuation in positioning error when the AUV’s motion status changed rapidly. It can be seen from Figure 5 and Table 1 that, when the AUV’s state of motion suddenly changes at 1000 s and 1600 s, the fluctuation in positioning error of the classical Kalman filter is the largest. It will take a long time to converge. The strong tracking filter, the improved adaptive filter, and the filtering gain compensation-based adaptive filter are able to significantly improve the stability. In particular, the fluctuation in the positioning error of our proposed method is very small (around 0). Regarding the peak–peak positioning error value, it is 6.9751 m, 1.3195 m, 0.3509 m, and 0.2476 m for the classical filter, the strong tracking filter, the improved adaptive filter, and the filtering gain compensation-based adaptive filter, respectively. Compared to these methods, our proposed method improves the peak–peak positioning error value by 96.45%, 81.24%, and 29.44%, respectively. So, the stability of our proposed method is the highest. It can be seen that the improved adaptive filtering algorithm based on filtering gain compensation that was designed in this study can better cope with the sudden changes in the AUV’s state of motion and improve the reliability of the system. This is because the filtering gain compensation algorithm compensates for the system error variance matrix when the AUV’s motion state experiences abrupt changes, which helps it to track the state of the system and improve the system’s positioning accuracy and reliability.

### 4.3. Positioning Efficiency Evaluation

We evaluated the positioning efficiency in terms of the positioning simulation time under the aforementioned model of the AUV’s motion. Table 2 lists the positioning simulation time of these methods.

From Table 2, it can be seen that the proposed filtering gain compensation-based adaptive filter has the largest positioning simulation time among these methods, while that of the classical Kalman filter is the smallest. This is because the filtering gain compensation-based algorithm itself analyzes the state of motion of the system and compensates for the error caused by the sudden change in the AUV’s state of motion, so the positioning simulation time is greater than that of the other methods. So, the efficiency of the proposed method is lower than that of the other methods. However, it should be pointed out that the loss of positioning efficiency remains within an acceptable range relative to the improvement in accuracy and stability.

Based on the above simulation analysis, we can conclude that the filtering effect of the gain compensation-based adaptive filter is obviously better than that of the classical Kalman algorithm and the strong tracking filtering algorithm in the integrated navigation process. In the case of a complex state of motion, the filtering effect is better than that of the improved adaptive filtering algorithm.

## 5. Discussion

To improve the positioning accuracy and stability of AUVs, we present a filtering gain compensation-based adaptive filter for the SINS/DVL integrated positioning system. In contrast to previous studies, we utilized indirect filtering and an integrated positioning error model to model the positioning parameters of the system. We also considered the error characteristics of the navigation parameters, and used the filtering gain compensation-based adaptive filter to filter the positioning parameters of the SINS/DVL integrated system in order to obtain an accurate and stable positioning result. However, we found that its calculation complexity is higher than that of other filtering methods. In our future work, we will perform further research on the SINS/DVL integrated positioning system based on the filtering gain compensation-based adaptive filter, cooperate with the Deep Sea Center, and evaluate our presented method in a real underwater environment.

## 6. Conclusions

The SINS/DVL integrated navigation system’s positioning process has insufficient positioning accuracy due to the long distances and the complex task environment. In this paper, we presented an adaptive filtering method based on filtering gain compensation for the SINS/DVL integrated navigation positioning system. In this method, the indirect filtering method was selected, and an integrated navigation error model was utilized to model the navigation parameters of the system. Then, based on the error characteristics of the navigation parameters, the improved adaptive filtering method for filtering gain compensation proposed in this paper was adapted to filter the navigation parameters of the system and obtain information on errors in the navigation parameters. Finally, the navigation parameters of the system were solved by combining the information on errors in the navigation parameters to obtain high-precision navigation results and realize the accurate positioning of underwater objects.

Our simulation results demonstrate that, compared with the classical Kalman filter and the strong tracking Kalman filter, the filtering method proposed in this paper can effectively improve the positioning error and has higher positioning accuracy. This is due mainly to the impact of the abrupt changes in speed and course on the positioning accuracy. The proposed filtering gain compensation-based algorithm can quickly track the system state in real-time and reduce the speed of error accumulation, so as to ensure the high-precision positioning of the AUV under long distance conditions and meet the positioning accuracy requirements in a complex task environment.

## Figures and Tables

**Figure 1 sensors-19-04576-f001:**
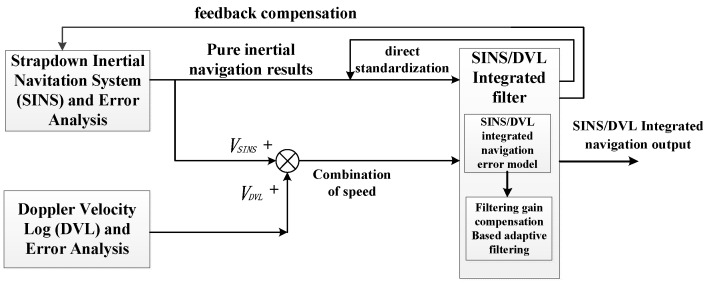
The framework for the strapdown inertial navigation system (SINS)/Doppler velocity log (DVL) integrated positioning system based on a filtering gain compensation adaptive Kalman filter.

**Figure 2 sensors-19-04576-f002:**
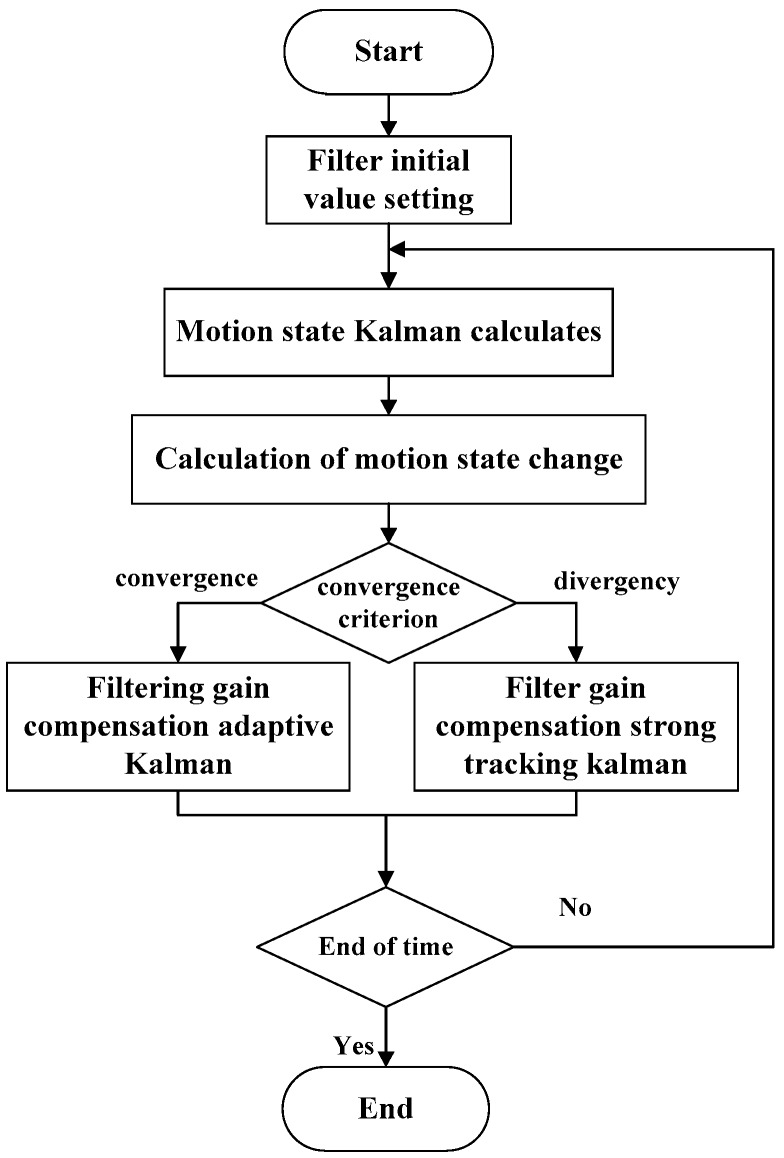
Workflow of the filtering gain compensation adaptive filter.

**Figure 3 sensors-19-04576-f003:**
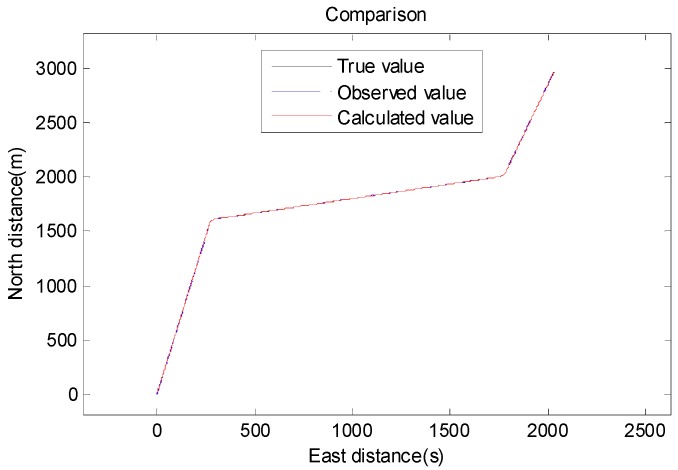
Schematic diagram of the autonomous underwater vehicle (AUV)’s motion.

**Figure 4 sensors-19-04576-f004:**
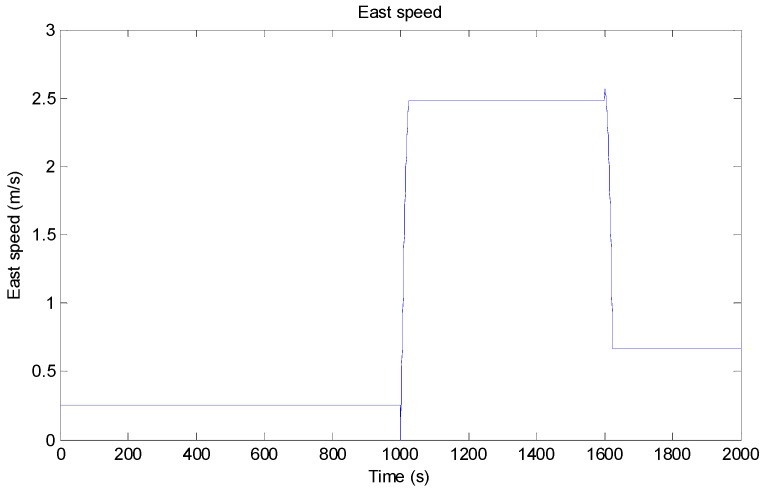
Schematic diagram of the AUV’s eastward speed.

**Figure 5 sensors-19-04576-f005:**
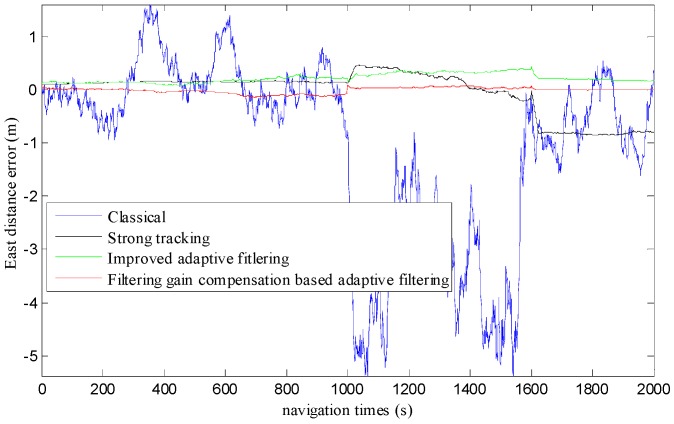
The positioning error of the different methods.

**Table 1 sensors-19-04576-t001:** The statistical positioning error parameter of the different methods.

Parameter	Classical	Strong Tracking	Improved Adaptive Filtering	Filtering Gain Compensation-Based Adaptive Filtering
Min (m)	–5.3766	–0.8611	0.0845	–0.1476
Max (m)	1.5984	0.4584	0.4353	0.1000
Peak–peak (m)	6.9751	1.3195	0.3509	0.2476
Mean (m)	–1.0731	–0.0342	0.2167	–0.0140
Improved	98.70%	59.06%	106.46%	-

**Table 2 sensors-19-04576-t002:** Comparison of the positioning simulation time of the filtering result.

Parameter	Classical	Strong Tracking	Improved Adaptive Filtering	Filtering Gain Compensation-Based Adaptive Filtering
Average positioning time (s)	0.40131	0.28413	0.3120	1.3474

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
