# Peer review of "A SINS/DVL Integrated Positioning System through Filtering Gain Compensation Adaptive Filtering"

_sensors, 2019, doi:10.3390/s19204576_

Round 1
Reviewer 1 Report
The paper investigate the adaptive kalman filter for gain compensation in the SINS/DVL integrated positioning system. The paper is good written and the reviewer read carefully and have following comments.
What is your main contribution? It is well known that the Kalmen filter was well studied, by using the method is not the novelty of the paper. The abstract and introduction section should be well present. In the simulation section, The figure 3 is totally unclear to valid your result. It should be imporved. is there some unstable problem in your adaptive kalman filter section? How to avoid it ?Author Response
Dear Editor-in-Chief and Reviewers,
We appreciate Editor-in-Chief for the helpful comments on our manuscript, and we would thank the reviewers for their constructive comments and suggestions. Now we are submitting the revised version after seriously taking into account the comments.
Sincerely,
Xiaozhen Yan, YiPeng Yang, Qinghua Luo, Yunsai Chen, Cong Hu.
Responses to the Editor-in-Chief's and the reviewers' comments are listed as follows.
Reviewer 1's comments:
Q1: What is your main contribution? It is well known that the Kalmen filter was well studied, by using the method is not the novelty of the paper.
A1: Thank you for your direction. Yes, The Kalman filter is a well-known processing method, and has been well studied. In our paper, different from previous research works, we make the following contributions.
1). For the fusion of positioning information, we present a general framework for an integrated positioning fusion system based on adaptive Kalman Filtering. It provides a general reference for information fusion methods.
2). we take full advantage of filtering gain compensation, strong tracking filtering and adaptive filtering to propose and utilize an organic combination of a filtering gain compensation adaptive filter and a filtering gain compensation strong tracking filter to fuse the positioning information to obtain higher accuracy and a more stable positioning result.
3). The proposed SINS/DVL integrated positioning system based on a filtering gain compensation adaptive filter can enhance underwater positioning performance and also provides reference value for other information fusion systems.
Q2: The abstract and introduction section should be well present.
A2: Thank you for your direction. We reorganized and revised the abstract and introduction section.
Revision: We made the following main revisions to make abstract and introduction section clear and readable.
1). In the fifth line of the Abstract section, we inserted the content of “and instability. To address this problem,” to indicate the challenges of underwater positioning system.
2). In the seventh line of the Abstract section, we inserted the content of “In the integrated positioning system, an organic combination of a filtering gain compensation adaptive filter and a filtering gain compensation strong tracking filter is explored to fuse position information to obtain higher accuracy and a more stable positioning result.” to illustrate the innovation of our work.
3). In the ninth line of the second paragraph of the Abstract section, we inserted the content of “, long working distance, random drift of the gyro, gradual divergence in the positioning error over time,” to indicate the positioning error resources.
4). In the eleventh line of the second paragraph of the Abstract section, we inserted the content of “Many research works have been conducted to improve underwater positioning performance. As the positioning error resources are not considered, these improvements do not show promise. In this paper, considering positioning error resources and their impact on the positioning result, we propose a SINS/DVL integrated positioning system based on filtering gain compensation adaptive filtering technology.” to describe the research works, changes and our innovation.
Q3: In the simulation section, the figure 3 is totally unclear to valid your result. It should be improved.
A3: Thank you for your direction. We will improve it.
Revision: We replaced the figure 3 and the figure 4 with a clear version, respectively.
Q4: is there some unstable problem in your adaptive kalman filter section? How to avoid it?
A4: Thank you for your direction. In our adaptive kalman filter, there are mainly two unstable factors in our proposed filtering gain compensation-based adaptive filter. One factor is the accumulation of positioning error during AUV’s motion. For this case, the adaptive filter could correct the positioning parameter on each step to minimize the accumulation of positioning error. The other factor is the sudden changes of the AUV’s motion state. As the correct ability of traditional adaptive filter is limited, to improve the stability, we take full advantage of filtering gain compensation, strong tracking filtering and adaptive filtering to propose and utilize an organic combination of a filtering gain compensation adaptive filter and a filtering gain compensation strong tracking filter to obtain higher accuracy and a more stable positioning result.
Additional revision 1: To make the paper clear, we organized and revised the theory and evaluation part. We check the whole manuscript carefully, revised and corrected the grammar, spelling, words and paragraph. We also turn to the MDPI professional English service to revise and polish our manuscript. All the revisions are highlighted in the upload version.
Additional revision 2:
1) To make the theory part more understandable, we organized and revised the subsection 3.42. Filtering gain compensation based adaptive Kalman filter to make it quite clear and readable. More specifically, we first reorganized subtitle 1) work flow of the filtering gain compensation adaptive filter. Then we reorganized subtitle 2) filtering gain compensation. Lastly we added subtitle 3) Adaptive Kalman filter and subtitle 4) Strong tracking Kalman filtering. We also added the corresponding content of the last two subtitles.
2) In subsection 4.1.3 Reference Positioning Methods, we organized the evaluation content to make it clear and readable, i.e., we listed three subtitle to illustrate the reference methods clearly: 1) Classical Kalman filter, 2) Strong tracking filter, 3) Improved adaptive filter.
3) In subsection 4.2.1 Positioning Accuracy Analysis, to make the evaluation part more understandable, we made a deep analysis on the positioning accuracy, i.e. we added two subsections 1). Comparison with classical Kalman filtering and 2). Comparison with strong tracking filtering and improved adaptive filtering to compare our proposed method with other three methods. We analyzed the accuracy difference and the corresponding mechanism.
Reviewer 2 Report
The paper is well organized and interesting. Although the verification setup is limited to simulation motion model and just one scenario, the results show some advantage of the presented positioning method over three other algorithm based on similar approach. There are also some language flaws, so the paper should be thoroughly revised by a native speaker.
However, there is a serious issue regarding the novelty of the paper. Quite recently the authors have published a paper [1] (Yang Y. P.; Yan X. Z.; Luo Q. H. A SINS/DVEL Integrated Navigation Positioning Method based on Improved Adaptive Filtering Technology) having almost the same title which thus is suspected to present the same algorithm. The authors have not resolved this doubt because the paper [1] is referenced only once as a general mention in the introduction. Since that paper is not available in IEEEXplore or any other accessible database yet, the authors should provide its copy to the reviewers in order to let them judge on the degree of novelty of the currently submitted manuscript. Also, the authors should make an explicit and more detailed reference to [1] and declare clearly the novelty of their current submission.
Until this I cannot proceed with more detailed review.
Author Response
Dear Editor-in-Chief and Reviewers,
We appreciate Editor-in-Chief for the helpful comments on our manuscript, and we would thank the reviewers for their constructive comments and suggestions. Now we are submitting the revised version after seriously taking into account the comments.
Sincerely,
Xiaozhen Yan, YiPeng Yang, Qinghua Luo, Yunsai Chen, Cong Hu.
Responses to the Editor-in-Chief's and the reviewers' comments are listed as follows.
Reviewer #2’s comments:
Q1: There are also some language flaws, so the paper should be thoroughly revised by a native speaker.
A1: Thank you for your direction. We have resorted to MDPI to revise and to polish our manuscript.
Revision: We check the whole manuscript carefully, revised and corrected the grammar, spelling, words and paragraph. We also turn to the MDPI professional English service to revise and polish our manuscript. All the revisions are highlighted in the upload version.
Q2: However, there is a serious issue regarding the novelty of the paper. Quite recently the authors have published a paper [1] (Yang Y. P.; Yan X. Z.; Luo Q. H. A SINS/DVEL Integrated Navigation Positioning Method based on Improved Adaptive Filtering Technology) having almost the same title which thus is suspected to present the same algorithm. The authors have not resolved this doubt because the paper [1] is referenced only once as a general mention in the introduction. Since that paper is not available in IEEEXplore or any other accessible database yet, the authors should provide its copy to the reviewers in order to let them judge on the degree of novelty of the currently submitted manuscript. Also, the authors should make an explicit and more detailed reference to [1] and declare clearly the novelty of their current submission.
A2: Thank you for your direction. Reference [1] is a conference paper, which appeared in the proceedings of SmartIOT 2019. The conference paper proposed a SINS/DVL integrated navigation positioning method based on improved adaptive filtering.
We extended it as a journal paper, which is recommended to the Sensors by the committee of the conference. We reorganized the paper to make it logical and readable, and we make a deep theory analysis on the filtering gain compensation based adaptive filter. In the Simulation Section, we make a comprehensive evaluation, including the positioning accuracy analysis, positioning stability analysis and positioning efficiency analysis. We also referenced Reference [1] as an important one.
Additional revision: we have made a comprehensive revision. We will make the novelty, theory and evaluation part more clear.
For the novelty part, we have organized and revised the Abstract section, Introduction section to render it clearer and readable.
For the theory part, in subsection 3.4.2, we have organized the content and added related content to make it theoretical.
For the evaluation part, in subsection 4.1.3, we have listed reference methods, and describe their characteristics. In subsection 4.2.1, we have added two subsections to analyze the poisoning accuracy deeply.
Revision: We have made the following main revisions to make it readable.
1). In the fifth line of Abstract section, we inserted the content of “and instability. To address this problem,” to indicate the challenges of underwater positioning system.
2). In the seventh line of Abstract section, we inserted the content of “In the integrated positioning system, an organic combination of a filtering gain compensation adaptive filter and a filtering gain compensation strong tracking filter is explored to fuse position information to obtain higher accuracy and a more stable positioning result.” to illustrate the innovation of our work.
3). In the ninth line of the second paragraph of the Abstract section, we inserted the content of “, long working distance, random drift of the gyro, gradual divergence in the positioning error over time,” to indicate the positioning error resources.
4). In the eleventh line of the second paragraph of the Abstract section, we inserted the content of “Many research works have been conducted to improve underwater positioning performance. As the positioning error resources are not considered, these improvements do not show promise. In this paper, considering positioning error resources and their impact on the positioning result, we propose a SINS/DVL integrated positioning system based on filtering gain compensation adaptive filtering technology.” to describe the research works, changes and our innovation.
5) To make the theory part more understandable, we organized and revised the subsection 3.42. Filtering gain compensation based adaptive Kalman filter to make it quite clear and readable. More specifically, we first reorganized subtitle 1) work flow of the filtering gain compensation adaptive filter. Then we reorganized subtitle 2) filtering gain compensation. Lastly we added subtitle 3) Adaptive Kalman filter and subtitle 4) Strong tracking Kalman filtering. We also added the corresponding content of the last two subtitles.
6) In subsection 4.1.3 Reference Positioning Methods, we organized the evaluation content to make it clear and readable, i.e., we listed three subtitle to illustrate the reference methods clearly: 1) Classical Kalman filter, 2) Strong tracking filter, 3) Improved adaptive filter.
7) In subsection 4.2.1 Positioning Accuracy Analysis, to make the evaluation part more understandable, we made a deep analysis on the positioning accuracy, i.e. we added two subsections 1). Comparison with classical Kalman filtering and 2). Comparison with strong tracking filtering and improved adaptive filtering to compare our proposed method with other three methods. We analyzed the accuracy difference and the corresponding mechanism.
8) We replaced figure 3 and figure 4 with a clear version, respectively.

Reviewer 3 Report
Unfortunately, in the reviewer's opinion, the novelty of the paper is too poor, both theoretically and experimentally. Consequently, the paper should not be published on the journal in the current form.
Author Response
Dear Editor-in-Chief and Reviewers,
We appreciate Editor-in-Chief for the helpful comments on our manuscript, and we would thank the reviewers for their constructive comments and suggestions. Now we are submitting the revised version after seriously taking into account the comments.
Sincerely,
Xiaozhen Yan, YiPeng Yang, Qinghua Luo, Yunsai Chen, Cong Hu.
Responses to the Editor-in-Chief's and the reviewers' comments are listed as follows.
Reviewer #3’s comments:
Q1: Unfortunately, in the reviewer's opinion, the novelty of the paper is too poor, both theoretically and experimentally. Consequently, the paper should not be published on the journal in the current form.
A1: Thank you for your direction. According to your comments, we have made a comprehensive revision. We will make the novelty, theory and evaluation part more clear.
For the novelty part, we have organized and revised the Abstract section, Introduction section to render it clearer and readable.
For the theory part, in subsection 3.4.2, we have organized the content and added related content to make it theoretical.
For the evaluation part, in subsection 4.1.3, we have listed reference methods, and describe their characteristics. In subsection 4.2.1, we have added two subsections to analyze the poisoning accuracy deeply.
Revision: We have made the following main revisions to make it readable.
1). In the fifth line of Abstract section, we inserted the content of “and instability. To address this problem,” to indicate the challenges of underwater positioning system.
2). In the seventh line of Abstract section, we inserted the content of “In the integrated positioning system, an organic combination of a filtering gain compensation adaptive filter and a filtering gain compensation strong tracking filter is explored to fuse position information to obtain higher accuracy and a more stable positioning result.” to illustrate the innovation of our work.
3). In the ninth line of the second paragraph of the Abstract section, we inserted the content of “, long working distance, random drift of the gyro, gradual divergence in the positioning error over time,” to indicate the positioning error resources.
4). In the eleventh line of the second paragraph of the Abstract section, we inserted the content of “Many research works have been conducted to improve underwater positioning performance. As the positioning error resources are not considered, these improvements do not show promise. In this paper, considering positioning error resources and their impact on the positioning result, we propose a SINS/DVL integrated positioning system based on filtering gain compensation adaptive filtering technology.” to describe the research works, changes and our innovation.
5) To make the theory part more understandable, we organized and revised the subsection 3.42. Filtering gain compensation based adaptive Kalman filter to make it quite clear and readable. More specifically, we first reorganized subtitle 1) work flow of the filtering gain compensation adaptive filter. Then we reorganized subtitle 2) filtering gain compensation. Lastly we added subtitle 3) Adaptive Kalman filter and subtitle 4) Strong tracking Kalman filtering. We also added the corresponding content of the last two subtitles.
6) In subsection 4.1.3 Reference Positioning Methods, we organized the evaluation content to make it clear and readable, i.e., we listed three subtitle to illustrate the reference methods clearly: 1) Classical Kalman filter, 2) Strong tracking filter, 3) Improved adaptive filter.
7) In subsection 4.2.1 Positioning Accuracy Analysis, to make the evaluation part more understandable, we made a deep analysis on the positioning accuracy, i.e. we added two subsections 1). Comparison with classical Kalman filtering and 2). Comparison with strong tracking filtering and improved adaptive filtering to compare our proposed method with other three methods. We analyzed the accuracy difference and the corresponding mechanism.
8) We replaced figure 3 and figure 4 with a clear version, respectively.
9) We check the whole manuscript carefully, revised and corrected the grammar, spelling, words and paragraph. The revisions are highlighted in the upload version.
10) To make the paper clear, we also turned to the MDPI professional English service to revise and polish our manuscript. The revisions are highlighted in the upload version.

Round 2
Reviewer 2 Report
As the authors admitted, the submitted manuscript is merely an extended version of the recently published conference paper. In this respect, the current paper does not propose any essential and relevant improvement nor wider experimental results over the previous one. If in the policy of the Sensor such incremental submissions acceptable and because the conference paper was recommended for publication in Sensors, I agree to the submitted paper for publication.
Author Response
Q1: English language and style are fine/minor spell check required.
A1: Thank you for your direction. We will check the whole paper, revise and correct English language gramma errors, spelling errors and other kinds of errors.
Revision: We made the following main revisions to make abstract and introduction section clear and readable. All the revisions are highlighted in the manuscript.
1). In many places in this paper, we revised the word “position” to “positioning” to make it readable.
2). In many places in this paper, we inserted the word “a” and “the” to make it readable.
3). In many places in this paper, we added the letter “s” at the end of many word.
4). In the title of section 3.4, we corrected the word “Adapitve” to “Adaptive”.
5). In the eleven page, we replaced the content “On the other hand” with “In addition” to make it clear.
6). In the eleven page, we corrected the content “can reduce to” with “can be reduced” to make it clear.
Additional revision 1: To make the Related Works part more understandable, in the first paragraph, we made a deep survey and inserted the discussion content of “The Kalman filter is a well-known powerful processing method, and has been well studied. In 998, Savkin et al presented a robust state estimation approach for a class of uncertain discrete-time systems with a deterministic description of noise and uncertainty. In this method, a recursive scheme was utilized to construct an ellipsoidal state estimation set of all states consistent with the measured output. And the result is promising [23]. In 2001, considering the uncertain parameters, literature [24] proposed a new approach to finite-horizon guaranteed state prediction for discrete-time systems affected by bounded noise and unknown-but-bounded parameter uncertainty. Wang F. et al presented an adaptive robust Kalman filtering algorithm. The algorithm explores a one-step predictor-corrector structure to minimize the mean square estimation error, so as to address the estimation problem the estimation problem that arises in linear time-varying systems with stochastic parametric uncertainties in 2002 [25]. To guarantee an optimized upper bound on the state estimation error covariance despite the model uncertainties as well as the linearization errors, Kai et al presented a novel robust extended Kalman filter (REKF) for discrete-time systems with stochastic uncertainties in 2009 [26].”. We also add the 4 references in the reference parts.
Additional revision 2: To make the Related Works part more understandable, in the first paragraph, we added the comparison content of “Different from previous works, in this paper, considering the parameter uncertainties and model uncertainties, we take full advantage of an organic fusion of strong tracking filtering and Sage–Husa adaptive filtering to achieve robust Kalman filtering for the SINS/DVL integrated positioning system. What is more important is that, we utilize an exponent filtering gain compensation strategy to quickly track the system state in real time, minimize the state estimation error and reduce the speed of the error accumulation. More specifically, when the initial motion state changes significantly, we increase the filter gain with an exponent compensation to enhance the effect of the new information according to the obtained observation value information. After the system is stabilized, we decrease the filter gain to ensure the optimal filtering of the system.” to make our contribution readable.

Reviewer 3 Report
I confirm the previous review.
Author Response
Dear Editor-in-Chief and Reviewers,
We appreciate Editor-in-Chief for the helpful comments on our manuscript, and we would thank the reviewers for their constructive comments and suggestions. Now we are submitting the revised version after seriously taking into account the comments.
Sincerely,
Xiaozhen Yan, YiPeng Yang, Qinghua Luo, Yunsai Chen, Cong Hu.
Responses to the Editor-in-Chief's and the reviewers' comments are listed as follows.
Reviewer #3’s comments:
Q1: I confirm the previous review.
A1: Thank you for your direction. We try to make a deep survey and make discussion with our proposed method.
Revision: We made the following main revisions to make Related Works section clear and readable. All the revisions are highlighted in the manuscript.
1) To make the Related Works part more understandable, in the first paragraph, we made a deep survey and inserted the discussion content of “The Kalman filter is a well-known powerful processing method, and has been well studied. In 998, Savkin et al presented a robust state estimation approach for a class of uncertain discrete-time systems with a deterministic description of noise and uncertainty. In this method, a recursive scheme was utilized to construct an ellipsoidal state estimation set of all states consistent with the measured output. And the result is promising [23]. In 2001, considering the uncertain parameters, literature [24] proposed a new approach to finite-horizon guaranteed state prediction for discrete-time systems affected by bounded noise and unknown-but-bounded parameter uncertainty. Wang F. et al presented an adaptive robust Kalman filtering algorithm. The algorithm explores a one-step predictor-corrector structure to minimize the mean square estimation error, so as to address the estimation problem the estimation problem that arises in linear time-varying systems with stochastic parametric uncertainties in 2002 [25]. To guarantee an optimized upper bound on the state estimation error covariance despite the model uncertainties as well as the linearization errors, Kai et al presented a novel robust extended Kalman filter (REKF) for discrete-time systems with stochastic uncertainties in 2009 [26].”. We also add the 4 references in the reference parts.
2) To make the Related Works part more understandable, in the first paragraph, we added the comparison content of “Different from previous works, in this paper, considering the parameter uncertainties and model uncertainties, we take full advantage of an organic fusion of strong tracking filtering and Sage–Husa adaptive filtering to achieve robust Kalman filtering for the SINS/DVL integrated positioning system. What is more important is that, we utilize an exponent filtering gain compensation strategy to quickly track the system state in real time, minimize the state estimation error and reduce the speed of the error accumulation. More specifically, when the initial motion state changes significantly, we increase the filter gain with an exponent compensation to enhance the effect of the new information according to the obtained observation value information. After the system is stabilized, we decrease the filter gain to ensure the optimal filtering of the system.” to make our contribution readable.
Additional Revision: We check the whole paper, revised and corrected English language gramma errors, spelling errors and other kinds of errors. All the revisions are highlighted in the manuscript.
1). In many places in this paper, we revised the word “position” to “positioning” to make it readable.
2). In many places in this paper, we inserted the word “a” and “the” to make it readable.
3). In many places in this paper, we added the letter “s” at the end of many words.
4). In the title of section 3.4, we corrected the word “Adapitve” to “Adaptive”.
5). In the eleven page, we replaced the content “On the other hand” with “In addition” to make it clear.
6). In the eleven page, we corrected the content “can reduce to” with “can be reduced” to make it clear.